# New Challenges in the Management of Cholangiocarcinoma: The Role of Liver Transplantation, Locoregional Therapies, and Systemic Therapy

**DOI:** 10.3390/cancers15041244

**Published:** 2023-02-15

**Authors:** Ezequiel Mauro, Joana Ferrer-Fàbrega, Tamara Sauri, Alexandre Soler, Amparo Cobo, Marta Burrel, Gemma Iserte, Alejandro Forner

**Affiliations:** 1Barcelona Clinic Liver Cancer (BCLC) Group, IDIBAPS, 08036 Barcelona, Spain; 2Centro de Investigación Biomédica en Red de Enfermedades Hepáticas y Digestivas (CIBERehd), Av. Monforte de Lemos, 3-5. Pabellón 11, Planta 0, 28029 Madrid, Spain; 3Hepatobiliopancreatic Surgery and Liver and Pancreatic Transplantation Unit, Department of Surgery, ICMDM, Hospital Clinic Barcelona, 08036 Barcelona, Spain; 4Faculty of Medicine, University of Barcelona, C/ de Casanova, 143, 08036 Barcelona, Spain; 5Medical Oncology Department, ICMHO, Hospital Clinic Barcelona, 08036 Barcelona, Spain; 6Radiology Department, CDI, Hospital Clinic Barcelona, 08036 Barcelona, Spain; 7Nuclear Medicine Department, CDI, Hospital Clinic Barcelona, 08036 Barcelona, Spain; 8Department of Interventional Radiology, CDI, Hospital Clinic Barcelona, 08036 Barcelona, Spain; 9Liver Unit, Liver Oncology Unit, ICMDM, Hospital Clinic Barcelona, 08036 Barcelona, Spain

**Keywords:** cholangiocarcinoma, liver transplant, locoregional therapies, systemic treatment

## Abstract

**Simple Summary:**

Cholangiocarcinoma (CCA) is a highly lethal neoplasia, which incidence has steadily increased in the last years. Although surgical resection remains the cornerstone treatment for CCA, complete resection is only achieved in one third of patients, and the risk of recurrence exceeds 60%, which impacts the long-term outcome. In this context, the use of other therapeutic strategies such as liver transplantation in selected candidates, locoregional treatments or new chemotherapy schemes based on immunotherapy or targeted therapies may contribute in improving the overall survival of patients with CCA. The development of new treatment strategies forces us to redouble collaborative efforts to conduct prospective, high-quality studies that shed light on their use and applicability. The purpose of this review is discussing the actual controversies and future perspectives in the management of CCA.

**Abstract:**

Cholangiocarcinoma (CCA) is a neoplasm with high mortality that represents 15% of all primary liver tumors. Its worldwide incidence is on the rise, and despite important advances in the knowledge of molecular mechanisms, diagnosis, and treatment, overall survival has not substantially improved in the last decade. Surgical resection remains the cornerstone therapy for CCA. Unfortunately, complete resection is only possible in less than 15–35% of cases, with a risk of recurrence greater than 60%. Liver transplantation (LT) has been postulated as an effective therapeutic strategy in those intrahepatic CCA (iCCA) smaller than 3 cm. However, the low rate of early diagnosis in non-resectable patients justifies the low applicability in clinical practice. The evidence regarding LT in locally advanced iCCA is scarce and based on small, retrospective, and, in most cases, single-center case series. In this setting, the response to neoadjuvant chemotherapy could be useful in identifying a subgroup of patients with biologically less aggressive tumors in whom LT may be successful. The results of LT in pCCA are promising, however, we need a very careful selection of patients and adequate experience in the transplant center. Locoregional therapies may be relevant in unresectable, liver-only CCA. In iCCA smaller than 2 cm, particularly those arising in patients with advanced chronic liver disease in whom resection or LT may not be feasible, thermal ablation may become a reliable alternative. The greatest advances in the management of CCA occur in systemic treatment. Immunotherapy associated with chemotherapy has emerged as the gold standard in the first-line treatment. Likewise, the most encouraging results have been obtained with targeted therapies, where the use of personalized treatments has shown high rates of objective and durable tumor response, with clear signs of survival benefit. In conclusion, the future of CCA treatment seems to be marked by the development of new treatment strategies but high-quality, prospective studies that shed light on their use and applicability are mandatory.

## 1. Introduction

Cholangiocarcinoma (CCA) is a highly lethal neoplasia comprising approximately 15% of all primary liver tumors. Its incidence is increasing worldwide, and despite significant advancements in the knowledge of molecular mechanisms, diagnosis, management, and survival have not substantially improved in the past decade [1,2]. These cancers are heterogeneous and are best classified according to the primary anatomic origin as intrahepatic CCA (iCCA), when located proximally to the second-order bile ducts within the liver parenchyma, perihilar CCA (pCCA), arising between the second-order bile ducts and the insertion of the cystic duct into the common bile duct, and distal CCA (dCCA), located in the common bile duct below the cystic duct insertion [1].

Several risk factors have been linked to CCA, most of them associated with chronic inflammation of the biliary epithelium and bile stasis. Some recognized risk factors such as obesity, metabolic syndrome, or high alcohol consumption have increased globally over recent decades, which could be contributing to increasing CCA incidence. However, the majority of CCA cases do not present any identifiable risk factors. 

In most cases, the diagnosis of CCA is established when the disease is already at advanced stages, which highly compromises access to effective treatment, resulting in a dismal outcome [3,4]. Therefore, prevention and early diagnosis remain the cornerstone for improving the survival of this devasting disease. 

Surgical resection is the best therapy for CCA [5,6]. Unfortunately, complete resection is possible in less than 15–35% of cases [4,7], and even in those patients in whom complete tumor removal is achieved, the risk of recurrence is greater than 60% [8,9]. Another radical option is liver transplantation (LT), but its use in CCA is controversial due to the high risk of recurrence and the lower survival benefit compared to other LT indications as well as the limited number of donors [5]. In addition, locoregional therapies have been recently proposed as a reliable treatment alternative for those patients with liver-only, unresectable CCA, but the low level of evidence supporting their efficacy impedes making any robust recommendation [10]. Finally, systemic therapy has rapidly evolved in the last years, and the irruption of targeted therapies and immunotherapy has changed the treatment approach. In this review, we will discuss the controversies in the therapeutical management of CCA.

## 2. Liver Transplantation in Cholangiocarcinoma

Theoretically, LT is an excellent treatment option for primary liver tumors due to (1) its capacity to completely remove the tumor (and the undetected liver micrometastasis), particularly when major resection is needed due to tumor extension/location, (2) its ability to eliminate the underlying chronic liver disease, and (3) the possibility of maximizing the survival benefit compared to alternative therapies. Regrettably, the major problem for the wide application of LT is the shortage of donors, since the number of candidates largely exceeds the available livers to be implanted. Due to the scarce number of donors, it is the usual policy to exclude from transplantation any patients with an expected suboptimal post-transplant survival (with a cutoff arbitrarily established to be at least greater than 50–60% at 5 years) [11,12]. In addition, the LT allocation policy should be adjusted to guarantee real access to LT, preventing drop-out due to tumor progression and withdrawal from the waiting list in an excessive proportion of patients, but at the same time, requiring an observation period which would allow for identifying biologically aggressive tumors that would be associated with a higher risk of unacceptable recurrence. 

### 2.1. Liver Transplantation in iCCA

iCCA was considered a contraindication for LT in most centers worldwide due to very poor initial results, i.e., a reported 2-year survival of around 30% [13,14]. These unacceptable outcomes were directly related to a high prevalence of microvascular invasion and poor tumor differentiation, particularly in patients with an unresectable or locally advanced tumor [15]. However, more recent retrospective studies have demonstrated encouraging results in terms of overall survival (OS) when a thorough selection of the population is performed. The relevance of selection based on tumor burden was demonstrated for the first time in an international multicenter study that included 48 patients who underwent LT and had been diagnosed with incidental iCCA in the explant. A total of 15 patients had “very early” iCCA (single tumor ≤ 2 cm) and 33 patients had “advanced” iCCA (single tumor > 2 cm or multifocal disease). After a median follow-up of 35 months, the 1-year, 3-year, and 5-year cumulative risks of recurrence were, 7%, 18%, and 18%, respectively, in the very early iCCA group vs. 30%, 47%, and 61% in the advanced iCCA group (*p* < 0.01). The 1-year, 3-year, and 5-year overall survival rates were 93%, 84%, and 65% in the very early iCCA group vs. 79%, 50%, and 45% in the advanced iCCA group (*p* < 0.02) [16]. Microvascular invasion and poor differentiation were associated with tumor recurrence in the multivariate analysis. Patients in the advanced iCCA group were divided into an intermediate stage (*n* = 6; single tumors 2.1–3 cm, not poorly differentiated) and an advanced stage (*n* = 27; all other tumors). The 1-year, 3-year, and 5-year overall survival rates were, 82%, 61%, and 61%, respectively, in the intermediate stage vs. 55%, 47%, and 42% in the advanced stage (*p* < 0.03) [16]. Ziogas et al. performed a meta-analysis of 18 studies, finding that the 5-year OS for very early iCCA was 71%, versus only 48% for advanced iCCA (single tumor > 2 cm or multiple tumors) [17]. Disappointingly, all these data come from retrospective studies, including mostly LT patients in whom iCCA was an incidental finding. Accordingly, prospective studies with well-defined inclusion and exclusion criteria and a predefined post-LT imaging follow-up are fervently needed. Furthermore, the outcome of LT should be analyzed according to the intention to treat the principle instead of only analyzing those patients finally transplanted. A multicenter, observational study (NCT02878473) aimed to prospectively evaluate the effectiveness of LT for very early iCCA is still ongoing. 

### 2.2. Locally Advanced, Unresectable iCCA and Liver Transplantation: Role of Neoadjuvant Therapy

iCCA patients with a stable, liver-limited disease on neoadjuvant therapy may have a favorable disease biology, with long-term survival after LT. Disappointingly, the evidence is scarce, and most studies are single-center, retrospective, and include a low number of patients and heterogeneous population in terms of tumor stage and neoadjuvant approach. The most relevant study, which is also performed within a study protocol at MD Anderson in Texas, was recently published by McMillan et al. [18]. Patients with CCA liver-only with the absence of vascular or lymph node involvement were considered for LT. Neoadjuvant therapy consisted of the first-line use of gemcitabine plus cisplatin (GemCis), and disease stability was required by radiological evaluation for at least six months. The treatments performed prior to LT in addition to GemCis were heterogeneous, including various types of locoregional therapies, liver resection, and targeted therapies such as inhibitors of isocitrate dehydrogenase 1 (IDH-1), fibroblast growth factor receptor (FGFR), and poly ADP-ribose polymerase (PARP). Over 11 years, 65 patients were evaluated, of whom 28 were denied for listing. Five patients were excluded after being eligible for resection due to tumor regression after neoadjuvant therapy. At the end of the follow-up, 18 out of 32 patients underwent LT and 14 did not (7 were still on the waiting list and 7 were because of tumor progression or death while on the WL). The time range between diagnosis and inclusion in the WL was very wide (74–1054 days), which could implicitly suggest a selection of patients linked to tumor biology. Intent-to-treat (ITT) survival analysis at 1, 3, and 5 years was 90%, 61%, and 49%, respectively. The recurrence-free survival (RFS) at 3 years was 52%, and 7 out of 18 transplant patients (39%) developed tumor recurrence (4 during the first year post-LT). In another retrospective analysis recently published by Ito T. et al. [19], 30 patients who underwent neoadjuvant therapy were finally transplanted. In this series, the neoadjuvant protocol was less defined, and the 5-year overall survival was 49%.

In conclusion, data on LT for unresectable iCCA are scarce and the level of evidence is low (Table 1). The response to neoadjuvant chemotherapy, especially in the context of new personalized target therapies, could identify patients with biologically less aggressive tumors in whom LT may offer long-term results. Prospective studies, with well-characterized and homogeneous populations in terms of baseline tumor burden, with clear-cut multimodal neoadjuvant protocols (locoregional and/or systemic treatment), and relevant outcomes (OS by ITT, RFS, and cancer-related survival), are mandatory to establish the role of LT in patients with locally advanced iCCA.

### 2.3. Perihiliar CCA (pCCA) and Liver Transplantation

The prognosis of pCCA is marked by frequent late diagnosis, which precludes the use of potentially curative treatments. The pCCA could develop in the context of primary sclerosing cholangitis (PSC) or de novo, in the absence of liver disease [1,4].

The indication of LT after neoadjuvant chemoradiation has been established as a therapeutic option with acceptable long-term OS results (>50% at 5 years) in carefully selected patients with early-stage unresectable pCCA and patients with pCCA associated with PSC [20]. Among different neoadjuvant chemoradiation strategies, the Mayo Clinic protocol based on strict criteria for diagnosis and patient selection, aggressive neoadjuvant chemoradiation, and surgical staging before transplantation, has been positioned as the best strategy. Tan et al. recently reported the Mayo Clinic results from 1993 to 2018. A total of 349 patients were initially assessed, but 277 (79%) underwent staging work-up, and in 60% (*n* = 211) LT was performed. According to ITT analysis (from the start of neoadjuvant therapy, including patients who did not undergo LT), the survival rates at 1, 5, and 10 years were 80%, 51%, and 46%, respectively, while the survival rates in those finally transplanted were 91%, 69%, and 62% [20]. The outcome in pCCA associated with PSC was significantly better than those arising within the healthy liver (5-year survival of 60% vs. 39%, respectively) [20,21], and the center experience positively impacts LT outcomes in patients with pCCA [22]. Other centers have reported poorer survival rates, inferior to 40% at 5 years [23,24,25,26,27], partially explained by the lower proportion of PSC-related pCCA and the lesser center experience. The use of living donor liver transplantation (LDLT) has been postulated as an interesting option since it does not directly impact the principles of allocation justice related to cadaveric LT. Regrettably, robust data comparing both options are scarce, but LDLT for de novo pCCA seems to be associated with higher disease recurrence and slightly worse OS [20].

Patient selection and neoadjuvant chemoradiation protocol are critical, being the Mayo Clinic proposal the most frequently evaluated. The Mayo Clinic protocol requires having a lesion with a radial diameter (perpendicular to the duct) ≤3 cm and without extension below the cystic duct. Endoscopic ultrasound-guided aspiration of regional hepatic lymph nodes is routinely performed prior to neoadjuvant therapy, and the presence of lymph node metastases is an exclusion criterion. Diagnostic biopsies, whether transgastric endoscopic or percutaneous transhepatic, are usually dismissed given the potential risk of seeding metastases in the peritoneum [21,28]. Vascular encapsulation and tumor extension along the duct, although not considered contraindications for neoadjuvant treatment, are conditions of a worse prognosis in terms of response [29]. The neoadjuvant therapy includes external beam radiation plus concomitant 5-fluorouracil and brachytherapy, followed by maintenance capecitabine until LT. After completion of neoadjuvant chemoradiation, patients should undergo staging laparoscopy prior to LT, comprising a complete examination of the abdominal cavity, routine biopsy of the regional lymph nodes, and biopsy of any other suspicious lesions. The timing of the staging surgery is a subject of debate, especially in PSC patients with advanced liver disease and complications associated with the presence of portal hypertension. However, the drop-out probability in patients with pCCA-PSC is usually lower than in patients with de novo pCCA (15% vs. 28%) [30]. In addition, the treatment of clinical complications during the neoadjuvant protocol is the cornerstone for LT outcome. Sarcopenia is frequently present in pCCA patients. It has shown an impact on the post-LT outcome and should be actively treated with nutritional support, which in some cases may include nasogastric or nasojejunal tube insertion for feeding [31]. Recurrent cholangitis because of biliary obstruction, biliary stenting, radiation-induced ductal injury, and/or underlying PSC has led to the development of antimicrobial resistance which increases the risk of intra-abdominal infections after LT. Finally, radiotherapy during the neoadjuvant protocol increases the risk of hepatic artery thrombosis and radiation-induced fibrosis [32,33].

Finally, recent publications found that LT in patients who meet the criteria for liver resection had better survival than that observed after resection, even when sub-analysis stratified by PSC was performed [34,35]. However, the survival benefit decreases when analyzed according to the ITT principle, which clearly calls into question the possibility of using donor livers for resectable pCCA [20,36]. An ongoing randomized, ITT multi-center trial in France TRANSPHIL (NCT02232932) comparing neoadjuvant chemoradiation and LT vs. upfront surgical resection will clarify this controversial topic.

## 3. Locoregional Therapies

Locoregional therapies (LRT) applicable to the treatment of iCCA include thermal ablation (TA), chemoembolization (TACE), radioembolization (TARE), chemotherapy hepatic arterial infusion (HAI), and external beam radiotherapy (EBRT). They have been postulated as an alternative to systemic therapy in those patients with liver-only, unresectable CCA, and in some cases, as neoadjuvant therapy prior to surgical resection/transplantation or as rescue therapy after systemic treatment failure.

### 3.1. Thermal Ablation

Although surgical resection is potentially the best therapeutic option in iCCA, some patients have liver-only disease categorized as unresectable due to localization and/or the presence of underlying cirrhosis with clinically significant portal hypertension (CSPH) and/or liver dysfunction that precludes liver resection. In this setting, TA might be a safe and effective treatment option. Unfortunately, the evidence to recommend these treatments is scarce and relies on retrospective studies, and in most cases, they are single-center and include a limited number of patients [37,38]. TA (with radiofrequency and microwave ablation being the most common techniques) is able to achieve local control of small iCCA lesions, focal and unresectable iCCA (either due to inadequate localization or CSPH), although its efficacy in terms of tumor response and survival outcomes are inferior to those obtained in the field of hepatocellular carcinoma (HCC) [39,40]. In a recent systematic review and pooled analysis, which included 15 studies with a total of 645 patients, mostly retrospective and monocentric, TA showed a pooled complete response rate of 93.9%, and a mean OS of 30.2 months (95% CI: 21.8–38.6) [10]. Noticeably, in more than 50% of the cases analyzed, TA was applied after post-resection recurrence, and 30% of the patients had liver cirrhosis. In patients with underlying cirrhosis, TA is a reliable treatment approach, and in those cases with single iCCA < 2 cm, TA obtained a similar survival to that obtained in HCC and comparable to that reported after surgical resection [41]. In summary, TA is feasible, safe, and may be a good alternative in selected unresectable patients.

### 3.2. Transarterial Chemoembolization, Radioembolization, and Chemotherapy Hepatic Arterial Infusion

Transarterial chemoembolization (TACE) has been evaluated in retrospective studies including a small number of patients with a very heterogeneous clinical profile, which makes it difficult to establish any recommendation. In a recent systematic review and pooled analysis, which included 22 studies with a total of 1145 patients, mostly retrospective and monocentric, TACE was associated with a pooled response rate of 23.4%, a mean PFS of 15 months, and OS of 15.9 months [10]. Moreover, the addition of TACE using irinotecan-loaded drug-eluting microspheres to GemCis vs. GemCis alone was tested in a small (*n* = 48) randomized controlled trial, showing a significant improvement in downsizing to resection (25% vs. 8%, *p* < 0.005) and an improved PFS and OS (33.7 vs. 12.6 months, *p* = 0.048), with an adequate safety profile [42]. Confirmatory, larger studies are needed before supporting this treatment combination.

Transarterial radioembolization (TARE) has also been evaluated in iCCA. However, most studies were single-center, including a small number of patients with heterogeneous inclusion criteria [43,44,45,46,47,48,49,50]. In a recent systematic review and meta-analysis including a total of 921 patients from 21 studies, TARE showed an overall disease-control rate of 82.3%, a median PFS and OS of 7.8 months and 12.7 months, respectively, and in 11% of the cases, patients were downstaged to being surgically resectable [51]. However, the high heterogeneity hampers data reproducibility. In addition, TARE was evaluated as associated with GemCis in a phase 2 study that included 41 naïve patients. The response rate and disease control rate according to RECIST were 41% and 98%, respectively. After a median follow-up of 36 months, the median PFS was 14 months (95% CI, 8–17 months), the median OS was 22 months (95% CI, 14–52 months), and nine patients (22%) could be downstaged to surgical resection, achieving R0 surgical resection in eight cases [52]. In addition, TARE was compared with systemic therapy (GemCis) in patients with locally advanced iCCA with liver-only disease (SIRCCA phase 3 trial, NCT02807181). Disappointingly, the study was prematurely interrupted because of low recruitment and its preliminary results have not yet been reported.

Finally, the efficacy and safety of chemotherapy hepatic arterial infusion (HAI) have been evaluated in small series including a heterogeneous population. In a recent systematic review, 331 patients from 16 studies were identified, many of them with bilobar involvement (75%), multifocality (66%), and a high percentage of macrovascular invasion (~40%). HAI showed a pooled response rate of 41.3%, and a PFS and OS of 10 months and 21.3 months, respectively [10].

### 3.3. External Beam Radiotherapy

The role of EBRT in the treatment of iCCA is also uncertain. Only one study was reported as prospective [53], but most patients were already treated with chemotherapy. In a systematic review and meta-analysis that included 541 patients, the 2-year local control rate, PFS, and OS were 69.1%, 15.6 months, and 18.9 months, respectively [10]. In addition, a recent registry study from the United States of America reported that the use of high-dose/ablative radiotherapy (>85 Gy) was associated with an improved outcome compared to conventional doses [54], which suggests that ERBT may be effective in some selected cases.

In a summary, locoregional procedures are safe and have shown some signals of efficacy and might be considered an alternative to systemic therapy in selected unresectable patients with iCCA.

## 4. Systemic Therapy

Treatment for patients with locally advanced and/or metastatic disease relies on the use of systemic therapy. Until recently, the only option that demonstrated survival benefit at the first line in advanced patients ineligible for surgical or locoregional options was the combination of GemCis. The pivotal ABC-02 study demonstrated a median overall survival (mOS) of 11.7 months (95% CI, 9.5 to 14.3) for GemCis compared with 8.1 months (95% CI, 7.1 to 8.7) for gemcitabine alone (HR, 0.64; 95% CI, 0.52 to 0.80; *p* < 0.001) [55]. The best results of GemCis were obtained in patients with iCCA, liver-only involvement [56]. Other combinations such as FOLFIRINOX (5-fluorouracil, oxaliplatin, and irinotecan) did not improve PFS compared with GemCis [57], and gemcitabine plus S-1 (an oral combination of the 5-fluorouracil prodrug tegafur with gimeracil and oteracil) demonstrated non-inferiority to gemcitabine and cisplatin in a randomized phase 3 trial [58]. Very recently, an open-label, non-inferiority phase 3 trial showed that capecitabine plus oxaliplatin (XELOX) was not inferior to gemcitabine plus oxaliplatin (GEMOX) in terms of 6-months PFS rate (46.7% [95% CI 41.5–51.8] vs. 44.6% [95% CI 39.7–49.3]) and thus XELOX may be an alternative to GEMOX in first-line setting [59]. Another promising combination is NUC-1031, a phosphoramidate transformation of gemcitabine, combined with cisplatin, which showed a promising objective response rate (ORR) (63.6% in the efficacy-evaluable population) in the ABC-08 phase Ib study [60]. This combination is currently under evaluation in a phase 3 trial in which patients are being randomized to NUC-1031 combined with cisplatin or GemCis (Nutide-121 trial; NCT04163900). Finally, results are also awaited from a phase II/III, multicenter, randomized, placebo-controlled study of GemCis with or without Bintrafusp Alfa (M7824) as the first-line treatment of BTC (NCT 04066491).

Immunotherapy has significantly expanded the scope of cancer treatment in recent years and its role in CCA is extensively revised elsewhere [61]. The most promising results have come from the phase 3 randomized, double-blind, placebo-controlled TOPAZ-01 trial, which demonstrated that durvalumab (PDL-1 antibody) plus GemCis significantly improved survival compared to GemCis plus placebo in advanced biliary tract cancer (BTC) [62]. Patients in the experimental arm received 1500 mg of durvalumab every 3 weeks with GemCis for up to eight cycles, followed by durvalumab 1500 mg every 4 weeks, until disease progression or unacceptable toxicity. 

The mOS was 12.8 months vs. 11.5 months (HR, 0.80; 95%CI, 0.66–0.97; *p* = 0.021), the median PFS (mPFS) was 7.2 months vs. 5.7 months (HR, 0.75; 95%CI, 0.64–0.89; *p* = 0.001), and ORR was 26.7% vs. 18.7% in durvalumab plus GemCis vs. GemCis plus placebo, respectively. The combination of durvalumab and GemCis was well tolerated, and grade 3 or 4 treatment-related adverse event rates were similar between both groups (62.7% with durvalumab vs. 64.9% with placebo). Updated OS and safety data after 6.5 months of additional follow-up have been recently reported in ESMO 2022. When compared with placebo, the addition of durvalumab to GemCis resulted in a longer median OS of 12.9 (11.6–14.1) months versus 11.3 (10.1–12.5) months, respectively, HR 0.76 (95% CI 0.64–0.91), together with manageable safety [63]. Based on these positive results, durvalumab plus GemCis has become the new standard first-line systemic therapy option for advanced BTC.

More recently, the addition of tremelimumab (two-dosing regimen) to durvalumab plus GemCis vs. Gemcitabine +/-cisplatin did not add any substantial benefit in a multicenter, German phase 2 trial [64]. Finally, another promising combination is Pembrolizumab plus GemCis (NCT 04003636-KEYNOTE 966), whose results have recently been announced as positive in 1st line treatment.

In the second line setting, the addition of liposomal irinotecan to 5-fluorouracil and leucovorin significantly improved PFS (3.9 months [95% CI, 2.6–4.7] vs. 1.5 months [1.2–1.9]) compared with 5-fluorouracil and leucovorin; HR = 0.38 [0.26–0.54], *p* < 0.0001) in a multicenter, open-label, randomized, phase 2b (NIFTY) study in patients with BTC who progressed on first-line GemCis [65,66]. More recently, the ABC-06 study demonstrated the benefit of leucovorin, 5- fluorouracil, and oxaliplatin (FOLFOX) in the second-line setting compared with the placebo [67]; mOS was significantly longer in the FOLFOX group than in the placebo group (6.2 vs. 5.3 months; HR 0.69, 95%CI 0.50–0.97, *p* = 0·031). Based on these findings, FOLFOX should become standard-of-care chemotherapy in the second-line treatment for advanced BTC and the reference regimen for further clinical trials. Multiple case reports exist of patients with mismatch repair deficient (dMMR)/microsatellite instability-high (MSI-H) CCA treated with the PD-1 antibody pembrolizumab with promising ORR [68,69]. The phase 2 KEYNOTE-158 basket trial of pembrolizumab for previously treated MSI-H cancer included 22 patients with CCA, for whom complete response (CR) and partial response (PR) were achieved in 3 (13.6%) and 6 (27.3%) patients, respectively, with a median {range} of the duration of response of 30.6 {6.2 to 40.5}, and a mOS of 19.4 months (95% CI 6.5—not reached) [70,71]. Based on those results, the FDA approved Pembrolizumab as a second-line therapy for patients with MSI-H cancers which have progressed through prior therapy.

### Targeted Therapy in CCA

Relevant progress has been accomplished in the last years on the molecular biology of CCA, and related target therapies. Molecularly, CCA is a highly heterogeneous disease, with genomic differences between intra and extrahepatic CCA [1,2,72,73], with IDH and FGFR pathway alterations predominantly found in intrahepatic cases, along with RAS and ARID1A [74]. The European Society for Medical Oncology (ESMO) scale for the clinical actionability of molecular targets (ESCAT) [75] is present in around 50–60% of cases, including kinases (FGFR1/2/3, PIK3CA, ALK, EGFR, ERBB2, BRAF, and KRAS), other oncogenes (IDH1/2) and tumor-suppressor genes (BRCA1/2) [76]. The main results of targeted therapy in CCA are summarized in Table 2.

The recent discovery of FGFR2 fusions in patients with iCCA has been rapidly translated into a promising therapeutic target [77,78]. Pemigatinib, a selective inhibitor of FGFR1, 2, and 3, was tested in a multicenter, open-label, single-arm, multicohort, phase 2 study (FIGHT-202) in patients with previously treated, locally advanced or metastatic CCA with FGFR2 fusions or rearrangements [79]. A total of 146 patients were enrolled and 107 had FGFR2 fusions or rearrangements. The ORR was achieved in 38 (35.5%) patients (3 patients with complete responses) and the mOS was 21.8 (14.8-not estimated) months. The most frequent adverse events were hyperphosphatemia, alopecia, diarrhea, fatigue, and dysgeusia, but in only 9 % of cases was the treatment interrupted because of toxicity. Another selective, ATP-competitive inhibitor of FGFR, Infigratinib, was tested in a single arm, open-label phase 2 trial including 108 with FGFR2 fusions or rearrangements who were previously treated with at least one gemcitabine-containing regimen. The ORR was 23.1% (one complete response and 24 partial responses) with a safety profile similar to pemigatinib [80] In addition, Futibatinib, a highly selective, irreversible FGFR1–4 inhibitor, showed in an open-label phase 2 (FOENIX-CCA2) trial including 103 patients with an FGFR2 fusion/rearrangement, an ORR of 41.7% and 74% of responses which lasted ≥6 months. After a median follow-up of 25 months, the mature mOS was 20.0 months, with a 12-month OS rate of 73.1% and a similar safety profile [81]. Based on these results, the FDA and EMA granted accelerated approval to pemigatinib, infigratinib, and futibatinib for the treatment of adults with previously treated, unresectable locally advanced or metastatic CCA with FGFR2 fusion or other rearrangements. All these three agents are being currently tested by a large, phase 3 RCT in a first-line setting against GemCis (Pemigatinib: FIGHT 302, NCT: NCT03656536; Futibatinib: FOENIX-CCA3, NCT 04093362; infigratinib: NCT03773302).

In addition, IDH1 mutations occur in approximately 15% of patients with iCCA [1]. Ivosidenib (AG-120), is a targeted inhibitor of mutated IDH1 and its efficacy has been shown in a multicenter, randomized, double-blind, placebo-controlled, phase 3 study (CLarIDHy) including patients with previously treated, IDH1-mutant CCA. A total of 185 patients were randomly assigned (2:1) to oral ivosidenib (*n* = 124) or a matched placebo (*n* = 61). The placebo to ivosidenib crossover was allowed after radiological progression. PFS was significantly improved with ivosidenib compared with the placebo (2.7 vs. 1.4 months; HR 0.37, 95% CI 0.25–0.54; *p* < 0·0001) [82]. Median OS was 10.3 months with ivosidenib vs. 7.5 months with placebo (HR 0.79, 95% CI 0.56–1.12; *p* = 0.09), but when adjusted for crossover, mOS with placebo was 5.1 months (HR 0.49, 95% CI 0.34–0.70; *p* < 0.001) [83]. Based on those results, ivosidenib was recently approved by the FDA for chemotherapy-refractory, IDH1 mutated CCA. 

HER2 overexpression or amplification, which is present in 15% of all cases of biliary tract cancer, has been identified as a druggable molecular target. The safety and efficacy of anti-HER2 therapy have been tested in prospective phase I/II studies in first-line associated with GemCis [84] and in second-line in combination with mFOLFOX [85], showing promising activity with acceptable toxicity, warranting further investigation. Finally, the ROAR basket trial evaluated dabrafenib and trametinib in 43 patients with previously treated, BRAF^V600E^-mutated BTC. The independently evaluated ORR was 47% (95% CI 31–62), the mPFS was 9 months (95% CI 5–10), and the mOS was 14 months (95% CI 10–33) [86]. The FDA granted swift approval for the use of dabrafenib and trametinib for patients carrying the BRAF^V600E^ mutation and cancer progression after systemic therapy.

In light of this and other accumulating evidence that advanced BTCs are good candidates for molecular triage, ESMO recently recommended the routine use of next-generation sequencing to be performed in all CCA patients. Therefore, molecular analysis, preferably using whole-gene sequence platforms, should be carried out before or during first-line therapy to evaluate options for second and higher lines of treatments as early as possible in advanced disease [75,87].

## 5. Future Perspectives

Treatment of CCA is rapidly evolving. Figure 1 summarizes the grade of recommendation and level of evidence of the available treatment options in CCA. Despite the fact that CCA has been considered a contraindication for LT, the competitive results in selected patient populations after the application of neoadjuvant therapy, which may allow for the evaluation of tumor biology, forces us to reconsider the potential role that LT may have in these patients. In the future, tumor biology assessment will be a critical determinant for patients’ outcomes in the setting of LT, and tools such as liquid biopsy or the integration of genomic and radiological information may help to improve patient selection. Further prospective studies with strict inclusion criteria and homogeneous and clear-cut multimodal neoadjuvant protocols are mandatory to establish the role of LT in patients with CCA. 

In addition, locoregional therapies may have a role in unresectable iCCA. In those patients with single tumors smaller than 2 cm, particularly arising in patients with advanced chronic liver disease in whom resection may not be feasible, thermal ablation may become a reliable alternative. Intra-arterial procedures, particularly TARE, are under evaluation in patients with unresectable, liver-only disease as an alternative or in combination with systemic therapy.

Undoubtedly, the greatest advances in the management of CCA have occurred in systemic treatment. Immunotherapy has emerged as an effective treatment associated with chemotherapy, and ongoing trials are evaluating these agents in this orphan disease. The analysis of immune biomarkers will be critical for the selection of patients with a greater benefit from immunotherapy. However, we currently do not have these biomarkers to optimize patient selection. The most hopeful results have been obtained with targeted therapies. In CCA, actionable molecular alterations are found in nearly 50% of cases and several agents directed to those molecular disruptions have shown promising results, with high rates of ORR and durable tumor response and signals of survival benefit. The CCA is a good candidate for molecular triage. The need for adequate tumor tissue for molecular profiling and intra-patient tumor heterogeneity are often limitations when tissue is obtained by percutaneous biopsy or cytology. In this setting, a liquid biopsy may overcome this obstacle and, hopefully, will allow for the monitoring of clonal evolution during selective therapeutic pressure and the detection of residual molecular disease. Cell-free DNA (cfDNA) analysis is an attractive approach as it can provide genomic information when tissue cannot be obtained or is insufficient in quantity or quality, can better capture intra-patient tumor heterogeneity, and can even facilitate the study of the evolution and tumor resistance, with potential predictive and prognostic capacity. These potential benefits might overcome the limitations of the use of tissue for NGS analysis, where, in more than 25% of cases, the isolated DNA does not qualify for a reliable molecular analysis.

## 6. Conclusions

The development of new treatment strategies represents a challenge in the therapeutic approach of CCA. The precision medicine based on molecular profiling is a reality and it will guide the systemic treatment. In addition, the applicability of locoregional therapies or liver transplantation based on an adequate patients’ selection will be crucial for improving the outcome. For all these strategies, the multidisciplinary and collaborative management will be essential and further studies are needed to confirm their benefits.

## Figures and Tables

**Figure 1 cancers-15-01244-f001:**
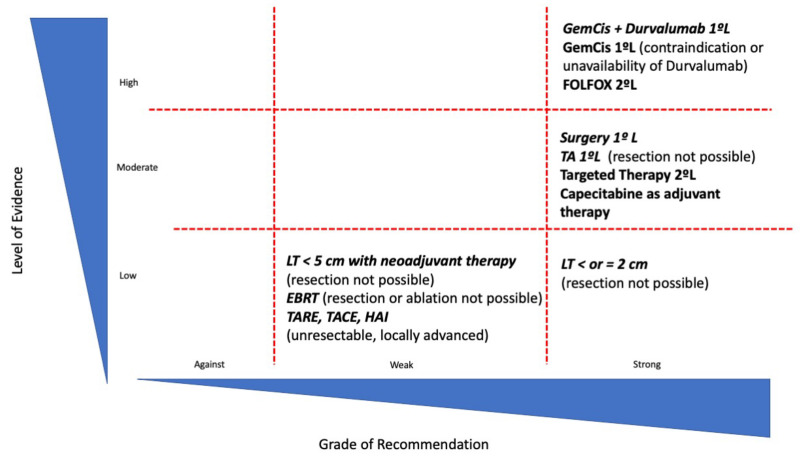
Representation of current treatments in CCA according to levels of evidence and strength of recommendation. TA: Thermal ablation; GemCis: Gemcitabine and Cisplatin; LT: Liver transplantation; EBRT: external beam radiotherapy; TACE: Transarterial chemoembolization; TARE: Transarterial radioembolization; and HAI: chemotherapy hepatic arterial infusion.

**Table 1 cancers-15-01244-t001:** Outcomes of LT in patients with unresectable iCCA.

Study	Design	Number of Patients	Neoadjuvant Therapy	Overall Survival
Sapisochin et al. (2016) [16]	Retrospective cohort multicenter. Incidental iCCA by pathological study.	48		1 year: 93% 3 years: 84% 5 years: 65%
McMillan et al. (2022) [18]	Prospective single-center case series.	18	Neoadjuvant chemotherapy (GemCis) and disease stability were required by radiological evaluation for at least six months. Treatments in addition to GemCis were heterogeneous: locoregional therapies, liver resection, and TT (IDH-1, FGFR, and PARP).	1 year: 100% 3 years: 71% 5 years: 57%
Ito et al. (2022) [19]	Retrospective, single-center, case series.	30	Neoadjuvant chemotherapy and or locoregional therapies.	1 year: 80% 3 years: 63% 5 years: 49%

Abbreviations: GemCis, Gemcitabine and Cisplatin; iCCA, intrahepatic cholangiocarcinoma; IDH, isocitrate dehydrogenase; FGFR, fibroblast growth factor receptor; and TT, targeted therapy.

**Table 2 cancers-15-01244-t002:** Main trials of targeted therapy in CCA.

Agent	Trial id and/or Name	Mechanism or Pathway	Phase	Study Population	Arms	Outcomes
**IDH mutations**						
Ivosidenib	NCT02989857ClarIDHy trial	IDH-1 inhibitor (decreases oncometabolite 2-HG)	3	Previously treated, advanced, IDH1-mutant CCA.	Ivosidenib vs. Placebo	mPFS (months): 2.7 (95% CI, 1.6–4.2) vs. 1.4 (1.4–1.6); HR: 0.37; (95% CI, 0.25–0.54) *p* < 0.0001 mOS (months): 10.3 (95% CI, 7.8–12.4) vs. 7.5 (95% CI, 4.8–11.1)
**FGFR alterations**						
Pemigatinib	NCT02924376 FIGHT-202	FGFR 1, 2, and 3 reversible inhibitors; FGFR fusions or rearrangements	2	Advanced, previously treated CCA with and without FGFR2 fusions/rearrangements/alterations.	FGFR2 rearrangements or fusion CCA Other FGF/FGFR alterations No FGF/FGFR alterations	ORR (%): 37 (95% CI, 27.9 –46.9) mOS (months): 17.5 (95% CI, 14.4–22.9) vs. 6.7 (95% CI, 2.1–10.5) vs. 4.0 (95% CI, 2.0–4.6)
Infigratinib (BGJ398)	NCT02150967PROOF-201	ATP-competitive FGFR 1, 2, and 3 tyrosine kinase reversible inhibitor	2	Locally advanced or metastatic CCA with FGFR2 fusions or rearrangements, previously treated with at least one gemcitabine-containing regimen.	Single arm	ORR (%): 23.1 (95% CI, 15.6–32.2)
Futibatinib (TAS-120)	NCT02052778FOENIX-CCA2	Highly selective, irreversible pan-FGFR antagonist	2	Advanced, previously treated iCCA with FGFR2 fusions/other rearrangements.	Single arm	ORR (%): 41.7 mPFS (months): 9 mOS (months): 21.7
Erdafitinib	NCT02699606LUC2001	Pan-FGFR kinase inhibitor	2	Patients previously treated, aCCA with FGFR alterations.	Single arm	ORR (%): 50.0
**HER2 alterations**						
Pertuzumab and trastuzumab	NCT02091141MyPathway	Monoclonal ab targeting HER2 domain II; monoclonal ab binds to domain IV of HER2	2	Previously treated, advanced BTC withHER2 amplification, overexpression, or both.	Single arm	ORR (%): 23 (95% CI, 11–39)
Neratinib	NCT01953926(SUMMIT trial)	Pan-HER irreversible TKI, with clinical activity against HER2	2	Previously treated, advanced BTC harboring HER2 somatic mutations.	Single arm	ORR (%): 12 (95% CI, 3–31) mPFS (months): 1.8 (95% CI, 1.1–3.7)
**BRAF V600E mutation**						
Dabrafenib and trametinib	NCT02034110ROAR trial	B-type Raf proto-oncogene, tyrosine kinase in the MAPK pathway	2	Previously treated, advanced BTC withBRAF V600E mutation.	Single arm	ORR (%): 47 (95% CI, 31 to 62)

Abbreviations: aCCA, advanced cholangiocarcinoma; BTC, biliary tract cancer; CCA, cholangiocarcinoma; HER2, human epidermal growth factor receptor 2; iCCA, intrahepatic cholangiocar-cinoma; IDH, isocitrate dehydrogenase; FGFR, fibroblast growth factor receptor; ORR, overall re-sponse rate; mOS, median overall survival; mPFS, median progression-free survival; and TKI, ty-rosine kinase inhibitor.

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
