# Peer review of "New Challenges in the Management of Cholangiocarcinoma: The Role of Liver Transplantation, Locoregional Therapies, and Systemic Therapy"

_cancers, 2023, doi:10.3390/cancers15041244_

Round 1

Reviewer 1 Report

The authors present a comprehensive narrative review on the current management of cholangiocarcinoma. The structure of their paper is based on the three main treatment approaches; namely liver transplantation, locoregional therapies and systemic treatment. Only a few comments:

1) Taking into consideration the available data and their level of evidence, I would start by discussing first (i) the systemic treatments eg surgery - indications, role of (neo)adjuvant treatment, salvage chemotherapy and targeted therapy, then locoregional treatment (ii), and, finally, liver transplantation (iii).

2) The authors should add a comment on when they suggest to perform the NGS in patients with cholangiocarcinoma. Should we perform a whole-genome analysis or targeted analysis on specific druggable targets and which?

3) In the last section on future directions, the authors discuss the role of liquid biopsies. I think they should add a paragraph discussing the available data, pros and cos, and their potential for use in clinical practice

Author Response

The authors present a comprehensive narrative review on the current management of cholangiocarcinoma. The structure of their paper is based on the three main treatment approaches; namely liver transplantation, locoregional therapies and systemic treatment. Only a few comments:

  • Taking into consideration the available data and their level of evidence, I would start by discussing first (i) the systemic treatments eg surgery - indications, role of (neo)adjuvant treatment, salvage chemotherapy and targeted therapy, then locoregional treatment (ii), and, finally, liver transplantation (iii).

Thank you very much for your comments. We have established the current order based on being able to emphasize the new therapeutic treatment challenges in different scenarios. In agreement with your comment, the longer section of this review is related to systemic therapy, where despite having the higher level of evidence, we find most uncertainness. In addition, the level of evidence is nicely summarized in figure 1.

  • The authors should add a comment on when they suggest to perform the NGS in patients with cholangiocarcinoma. Should we perform a whole-genome analysis or targeted analysis on specific druggable targets and which?

Thank you very much for your comments. The main guidelines encourage the use of whole-genome analysis, if available. We have added the data to the “Targeted therapy in CCA” section.

  • In the last section on future directions, the authors discuss the role of liquid biopsies. I think they should add a paragraph discussing the available data, pros and cos, and their potential for use in clinical practice.

Thank you for this important suggestion. We have added comments in the "Future Perspective" section

Reviewer 2 Report

The present Review regarding cholangiocarcinoma is rather a simple one. There many similar published in the literature in the last few months

Suggestions to become more uniques

1) Indicate treatment sequences depending on the location of cholangiocarcinoma

2) A table showing the pros and cons of the newly arrived treatment options

3) A more critical approach in the Discussion Session - What are the next steps? What are the scientific gaps?

Author Response

The present Review regarding cholangiocarcinoma is rather a simple one. There many similar published in the literature in the last few months

Suggestions to become more uniques

  • Indicate treatment sequences depending on the location of cholangiocarcinoma.

Thank you very much for your suggestion. The purpose of the review is to focus on new therapeutic challenges. In the case of systemic treatment, all trials demonstrating survival benefit included biliary track cancer (BTC), and the survival benefit was maintained in post-hoc analysis based on tumor localization. Accordingly, the treatment sequence does not vary according to localization, except the use of locoregional therapies that are feasible in intrahepatic localization. 

  • A table showing the pros and cons of the newly arrived treatment options

Thank you very much for your comment. For this purpose, we have already included both tables describing the indication of LT in different scenarios (table 1) and current evidence behind the different targeted therapies evaluated in CCA (table 2)

  • A more critical approach in the Discussion Session - What are the next steps? What are the scientific gaps?

Thank you very much for your suggestion. We have added comments about it in the “Future Perspective” section.

Reviewer 3 Report

The authors presented a very interesting review regarding the role of liver transplantation (LT) in the pathology of cholangiocarcinoma (CCA). The authors have composed their review concept very well and the subtitles are well defined. However, the authors take a look at the molecular behaviors of CCA, describing the re-interactions with their target therapies. The document is very easy to read, clear and well addressed.

Furthermore, we would like to suggest the following modification to the authors: The authors may attempt to construct an iconographic figure representing the structure of the HPB complex, indicating the anatomical differences regarding the different origins of the CCAs. The manuscript is eligible for publication.

Author Response

Thank you very much for your comments, they have been of great help to improve the manuscript

Round 2

Reviewer 1 Report

 Accept 

Reviewer 2 Report

Can be further published